# Selective Inhibition of HDAC Class I Sensitizes Leukemia and Neuroblastoma Cells to Anticancer Drugs

**DOI:** 10.3390/biomedicines9121846

**Published:** 2021-12-06

**Authors:** Elmira Vagapova, Maxim Kozlov, Timofey Lebedev, Karina Ivanenko, Olga Leonova, Vladimir Popenko, Pavel Spirin, Sergey Kochetkov, Vladimir Prassolov

**Affiliations:** 1Engelhardt Institute of Molecular Biology, Russian Academy of Sciences, Vavilov Street 32, 119991 Moscow, Russia; kozlovmavi@gmail.com (M.K.); lebedevtd@gmail.com (T.L.); karina.ivanenko@mail.ru (K.I.); leonova-kozma@mail.ru (O.L.); popenko@eimb.ru (V.P.); spirin.pvl@gmail.com (P.S.); kochet@eimb.ru (S.K.); prassolov45@mail.ru (V.P.); 2Center for Precision Genome Editing and Genetic Technologies for Biomedicine, Engelhardt Institute of Molecular Biology, Russian Academy of Sciences, Vavilov Street 32, 119991 Moscow, Russia

**Keywords:** histone deacetylases, leukemia, neuroblastoma, combinational therapy, receptor tyrosine kinases, venetoclax

## Abstract

The acquired resistance of neuroblastoma (NB) and leukemia cells to anticancer therapy remains the major challenge in the treatment of patients with these diseases. Although targeted therapy, such as receptor tyrosine kinase (RTK) inhibitors, has been introduced into clinical practice, its efficacy is limited to patients harboring mutant kinases. Through the analysis of transcriptomic data of 701 leukemia and NB patient samples and cell lines, we revealed that the expression of RTK, such as KIT, FLT3, AXL, FGFR3, and NTRK1, is linked with HDAC class I. Although HDAC inhibitors have antitumor activity, they also have high whole-body toxicity. We developed a novel belinostat derivative named hydrazostat, which targets HDAC class I with limited off-target effects. We compared the toxicity of these drugs within the panel of leukemia and NB cell lines. Next, we revealed that HDAC inhibition with hydrazostat reactivates *NTRK1*, *FGFR3*, *ROR2*, *KIT*, and *FLT3* expression. Based on this finding, we tested the efficacy of hydrazostat in combination with RTK inhibitor imatinib. Additionally, we show the ability of hydrazostat to enhance venetoclax-induced apoptosis. Thus, we reveal the connection between HDACs and RTK and describe a useful strategy to overcome the complications of single-agent therapies.

## 1. Introduction

Acute myeloid leukemia (AML) and neuroblastoma (NB) are highly heterogenic malignancies with poor outcomes despite advances in therapeutic regimens and considerable rates of remission. These types of childhood cancers share common features including the aberrant expression of receptor tyrosine kinases (RTK) such as KIT and NTRK1 [1]. RTKs are attractive targets for the development of inhibitors to treat various cancers including AML and NB. Although RTK inhibitors, especially FLT3 inhibitor-quizartinib, demonstrate single-agent efficacy by inducing complete remission in cancer patients, their utility is limited to FLT3-mutant AML patients [2,3]. Several RTKs, including the Trk family [4], KIT and PDGFR [5], and ALK [6], have been investigated as potential targets in NB patients; however, the evidence of their clinical significance was only limited [7,8,9]. Notably, RTK targeting has been hampered by acquired resistance across various cancers [10,11,12,13], which clearly illustrates the need for additional therapeutic targets in AML and NB patients. Chronic myeloid leukemia (CML) is another malignancy, where kinase inhibitors (imatinib and dasatinib) have shown their efficacy [14]; however, most patients in blast crisis fail to respond or quickly relapse [15].

Deregulation of the epigenetic apparatus that is essential for normal cell development is frequently observed in tumors, leading to the overexpression of oncogenes and silencing of tumor-suppressor genes. An important epigenetic mediator of transcription is histone acetylation by histone acetyltransferases (HATs) and deacetylation by histone deacetylases (HDACs). In total, two types of histone deacetylases are distinguished by their catalyst cofactors: zinc-dependent HDACs (classes I, II, and IV) [16] and NAD +-dependent HDACs (class III), also designated as SIRT (sirtuins) [17]. HDAC1–3 of class I are localized only in the nucleus of the cell while HDAC8 of class I and HDAC4, 5, 7, 9 of class IIA shuttle between the nucleus and cytoplasm, and finally, HDAC6, 10 of class IIB and HDAC11 of class IV function to a large extent in the cytoplasm [18]. Additionally, HDAC function in nuclei is not limited to histone modification, as these enzymes regulate gene expression as part of transcription elongation complexes [19]. Therefore, the aberrant activity of HDACs could drive tumor progression and resistance through the modulation of oncogene expression. In a variety of cancer types, HDAC function and/or expression is altered, and in the majority of cases associated with advanced disease and poor prognosis [20]. Thus, HDAC proteins are considered as relevant therapeutic targets in various tumors, which has driven the development of numerous HDACs inhibitors, such as belinostat, a broad-specificity inhibitor, that shows efficacy against T-cell lymphoma [21]. Pan-HDAC inhibitors often exhibit high toxicity and side effects, hence the need to identify the functions of specific HDACs and develop new specific HDAC inhibitors and therapy regimens. Additionally, accumulating data on the implication of individual HDACs in the development and progression of cancer including AML and NB have led to the development of more specific inhibitors. Several HDACs inhibitors demonstrate their anti-leukemia [22,23] and anti-NB [24,25] potency in disease models. Currently, several HDAC inhibitors are involved in clinical trials to treat leukemia and NB (NCT04326764, NCT03772925, and NCT02559778).

Furthermore, it was stated that several chemotherapeutic drugs can enhance HDAC class I expression; for example, doxorubicin and cytarabine induce HDAC3 expression in leukemia cells (K562 and THP-1), contributing to acquired resistance as HDAC3 is critical for DNA damage control [26,27,28]. Fusion proteins can recruit HDACs localized in the nucleus to the promotors of the oncogenes and tumor-suppressor genes, thus playing a key role in the differentiation and proliferation of cancer cells. For instance, chimeric proteins RUNX1-RUNX1T1 and PMLa-RARa form protein complexes with HDAC class I proteins. It is known that these aberrant complexes regulate the expression of genes including antiapoptotic protein BCL-2, which promotes cancer cell survival [29,30,31]. HDAC8 is overrepresented in the cancer cells of patients with advanced metastatic NB and is downregulated in INSS 4S NB (with a high rate of spontaneous regression) [32]. Whilst HDAC inhibitors are an emerging class of effective targeted anticancer agents, more work is required to determine how these agents might best be deployed to improve treatment outcomes for NB and leukemia patients. Recently, we developed the N’-propylhydrazide analog of belinostat, named in this work as hydrazostat, which has proven to be a potent HDAC class I inhibitor with limited off-target effects [33]. We decided to test its toxicity within a broad panel of leukemia and NB cell lines. Based on the observed differences in sensitivity that were cell origin independent, we aimed at the identification of HDAC signatures in leukemia and NB cancer cells. We show that RTK expression is linked with HDAC class I coding genes expression in leukemia and NB patient samples and cell lines. Furthermore, the dependency is beneficial for the effective suppression of cancer cell growth by dual targeting of RTK and HDAC class I. Revealed vulnerabilities are beneficial for the development of novel therapeutic approaches for NB and leukemia patients.

## 2. Materials and Methods

### 2.1. Cell Lines and Cellular Assays

Human *FLT3*-*ITD*^+^ cells (MV-4-11), *FLT3*-*ITD*^–^ and *KIT*-N822K^−^ cells (THP-1, HL-60, U-937, and K562), *MYCN*^+^ (LAN-1, Kelly, and SK-N-BE), and *MYCN*^−^ (SH-SY5Y, SK-N-AS, and SK-N-SH) cells were cultured in RPMI-1640 (Thermo Fisher Scientific, Gibco, Paisley, Renfrewshire, Scotland, UK) with 10% fetal bovine serum (FBS) (Thermo Fisher Scientific, Gibco, Paisley, Renfrewshire, Scotland, UK). KIT-N822K^+^ (Kasumi-1) cells were maintained in RPMI 1640 with 20% FBS. To assess cell viability, in vitro cells were cultured in the presence of the drugs for 72 h and manually counted in a Neubauer chamber with trypan-blue (Dia-m) exclusion. Percent of apoptotic cells was assessed by BD LSRFortessa Flow Cytometer (BD Biosciences, Franklin Lakes, NJ, USA)—cells were stained with Annexin V-FITC (Molecular Probes, ThermoFisher, Paisley, Renfrewshire, Scotland, UK) and propidium iodide (Sigma Aldrich, Saint Louis, MO, USA). Bliss synergy scores of compounds were calculated with SynergyFinder [34].

### 2.2. Drugs and Compounds

Hydrazostat (formerly called belinostat-PH) was synthetized according to our paper [33]. Other drugs were purchased from Selleckhem, Houston, TX, USA and Sigma Aldrich, Saint Louis, MO, USA, solved according to manufacture protocol in DMSO or sterile water, and stored at −20 °C. Details are provided in Appendix A.

### 2.3. GEO Datasets

Datasets GEO ID: GSE14468 (AML patients, *N* = 525) [35], GEO ID: GSE16476 (NB patients, *N* = 88) [36], and GEO ID: GSE36133 (leukemia and NB cell lines, *N* = 613) containing RNA-sequencing data were downloaded from R2: Genomics Analysis and Visualization Platform [37] and analyzed with Python. Code is available at CancerCellBiology, GitHub [38].

### 2.4. Elastic NET Regularization

To identify gene related between HDAC and RTK gene expression, we used data for 29 NB solid cancer and 59 leukemia cell lines from Genomic of Drug Sensitivity in Cancer (PMID: 23180760), and 525 AML (GSE14468) and 88 NB (NB patients) tumors. We applied linear regression with elastic net regularization to determine how the expression of individual RTK gene depends on HDAC genes expression:$$RTK=y_{o}\;+\;\underset{i}{\sum }\omega_{i}*exp_{i}$$
$$\hat{\omega }=\;\underset{\omega }{\min}(∥S-\;X_{\omega }{∥}_{2\;}+\;\alpha *l1*∥\omega {∥}_{1}+\alpha *\frac{\left(1-l1\right)}{2}*∥\omega {∥}_{2})\;$$
where *exp_i_*—expression of HDAC gene *i*, *ω_i_*—weight of gene *i*, *y_o_*—intercept, $\hat{\omega }$—weights estimator, RTK—expression of individual RTK gene, *X*—expression matrix, *ω*—weights matrix, and *α* and *l*1—elastic net penalty parameters. For linear regression and elastic net regularization, we used the scikit-learn Python library. Penalty parameters for elastic net regularization were determined by performing 10-fold cross validation for each dataset.

### 2.5. qRT-PCR

Total RNA was isolated with Trizol reagent (Invitrogen, Carlsbad, CA, USA), complementary DNA was synthesized with RevertAid First Strand cDNA Synthesis Kit (Thermo Fisher Scientific, Waltham, MA, USA), and SYBR Green (Evrogen, Moscow, Russia) quantitative polymerase chain reaction (PCR) was performed in triplicate and analyzed with CFX96 Touch Real-Time PCR Detection System (Bio-Rad, Hercules, CA, USA); primer sequences are listed in Appendix A. Cycle threshold values were normalized to endogenous glyceraldehyde-3-phosphate dehydrogenase (GAPDH).

### 2.6. Immunocytochemistry and Confocal Imaging

THP-1 and K562 cells were fixed with 4% formaldehyde solution (Sigma Aldrich, Saint Louis, MO, USA) in 0.1 M phosphate-buffered saline (PBS) (pH 7.3) for 15 min, washed with PBS (3 × 10 min), treated with 0.2% Triton X-100 in PBS (10 min at room temperature), washed with PBS (10 min), blocked with 1% BSA/22.52 mg/mL glycine/1% Tween in PBS. THP-1 cells were incubated overnight at 4 °C with anti-FLT3 antibody conjugated with FITC (ab183211, Abcam, Cambridge, UK) diluted 1:50 in PBS containing 1% BSA. K562 cells were incubated overnight with anti-KIT conjugated with FITC (ab119107, Abcam, Cambridge, UK) diluted 1:50 and anti-phosphoERK primary (ab4377 s, Abcam, Cambridge, UK) antibodies diluted 1:200 in PBS containing 1% BSA. After overnight incubation, K562 cells were washed with PBS and incubated at room temperature for 1 h with secondary antibodies conjugated with Alexa 647 (ab150075, Abcam, Cambridge, UK) diluted 1:500 in PBS containing 1% BSA. Finally, after three washes with PBS, THP-1 and K562 cells were mounted in Slowfade gold medium (s36936, Invitrogen, Waltham, MA, USA) containing 1 μg/mL DAPI (Sigma Aldrich, Saint Louis, MO, USA) and sealed with nail polish. Confocal 8-bit digital images were obtained using a Leica TCS SP5 laser-scanning microscope (Leica, Wetzlar, Germany) equipped with an HCX PLAPO CS 63 × 1.4 oil immersion lens. The image acquisition (sequential mode) parameters were as follows: DAPI fluorescence (DNA staining dye) with excitation at 405 nm and emission at 412–484 nm; FITC: excitation 488 nm, emission 495–560 nm; Alexa Fluor 647: excitation 633 nm, emission 440–560 nm. Images were processed using LAS AF Lite 4.0 software.

### 2.7. Quantification of ERK Activity with ERK-KTR Reporter

We used SH-SY5Y:ERK-KTR cell line established as described in [39] to measure ERK activity in live neuroblastoma cells. Briefly, ERK-KTR translocation reporter was introduced into SH-SY5Y cells through lentiviral transduction. For that purpose, we used pLentiCMV Puro DEST ERKKTRClover plasmid (Addgene plasmid #59150), which encoded ERK docking domain ELK, nuclei localization site and nuclei extraction site (containing phosphorylation sites) fused with fluorescent protein mClover [40]. Lentiviral particles were collected with the culture medium of HEK-293T cells 24 and 48 h after their transfection with plasmids coding structural elements of the lentivirus as described at Lentiviral Gene Ontology Vectors [41]), plasmid coding G protein (envelope) of the vesical stomatitis virus VSV and #59150 plasmid. Cells were imaged with Leica DMI8 automated microscope using 10× magnification lenses. Nuclei of the cells were stained with 500 ng/mL Hoechst-33342 for 30 min before imaging. For each drug/combination of drugs we imaged three wells and three fields in each well. Imaging of the cells was performed in two channels-461 nM (for Hoechst) and 520 nM (for mClover). Illumination correction, segmentation of nuclei and cells, calculation of cytoplasm to nucleus ratios of individual cells (C/N ratio) corresponding to ERK activity were made in CellProfiler.

### 2.8. Statistical Analysis

Statistical analysis was performed using GraphPad Prism Version 9.2.0 San Diego, CA, USA. Types of tests used are defined in the description of figures.

### 2.9. Code Availability

Code is available at CancerCellBiology, GitHub [38]. 

## 3. Results

### 3.1. Leukemia and Neuroblastoma Cells Have Different Sensitivity to Belinostat and Hydrazostat

We have shown that the *N*′-propylhydrazide analog of hydroxamic inhibitor belinostat (hydrazostat, previously described as belinostat-PH) is a potent HDAC class I inhibitor with negligible off-target activities across the other HDACs in human hepatoma cells [33]. Here, we conducted a comparative examination of the utility of belinostat and hydrazostat to suppress the viability of leukemia (AML and CML) and NB cells.

We began by assessing the toxicity of these two inhibitors on a panel of cancer cell lines: 6 leukemia (K562, Kasumi-1, HL-60, MV-4-11, THP-1, and U-937) and 6 NB (SH-SY5Y, Kelly, LAN-1, SK-N-AS, SK-N-SH, and SK-N-BE) cell lines. Both inhibitors efficiently suppressed cell growth in μM concentrations. The IC50 values for belinostat ranged from 0.9 μM (HL-60) to 5.1 μM (Kelly) and for hydrazostat from 0.7 μ M (HL-60) to 3.7 μM (SK-N-SH) (Figure 1a and Appendix A).

We observed two types of cellular response to HDAC inhibitors: equal toxicity of the components and increased sensitivity to hydrazostat. Notably, these effects did not depend on the cell origin. All in all, IC50 values for both drugs determined for SK-N-SH, U-937, MV-4-11, Kasumi-1, and SK-N-AS cells were comparable (*p*-value = 0.2944). Kelly, K562, THP-1, LAN-1, SH-SY5Y, and SK-N-BE cells were more sensitive to hydrazostat (*p*-value = 0.0011) (Figure 1b). Further, we aimed at the investigation of the mechanism of the sensitivity of leukemia and NB cells to HDAC inhibition.

### 3.2. The Expression of HDAC Class I Correlates with the Expression of Kinases NTRK1 and FGFR3 in Leukemia and NB Patients and Cell Lines

Although HDACs are known to act as direct and indirect epigenetic regulators of RTK signaling by affecting RTK effector gene expression and downstream signaling [42], the contribution of particular HDACs in the regulation of kinase coding genes in leukemia and NB cells is not completely understood. Thus, we aimed at the identification of HDAC signatures to find out RTK-based vulnerabilities of cancer cells.

First, we applied the elastic net regression model to determine the relation of HDAC class I and II coding genes with the genes coding human RTKs (Appendix A) using publicly available datasets of NB (GSE14468) and AML patients (GSE16476) where all genes coding HDAC1-8 were present. The elastic net model is frequently used to identify which gene expression contributes to a particular cancer-type-specific signaling pathway regulator [43]. Next, we calculated the SCORE and SCORE I to range the dependency of RTKs gene expression on *HDACs* expression (Figure 2a and Appendix A).

The analysis resulted in different top 15 scoring kinases for AML and NB patients. TIE1, KIT, NTRK1, IGF1R, PTK7, LTK, FGFR3, CSF1R, DDR2, AXL, TEK, INSRR, ROR1, AATK, and FLT3 for AML; NTRK1, PDGFRA, STYK1, INSRR, FLT4, TIE1, ROR2, ROS1, NTRK2, FGFR2, FGFR3, CSF1R, PDGFRB, ALK, and KIT for NB. Notably, TIE1, KIT, NTRK1, FGFR3, CSF1R, and INSRR were shared by both cancers. We have applied the same method to RNA-sequencing data of NB and leukemia (AML, ALL and CML) cell lines from Genomics of Drug Sensitivity in Cancer database (deposited at cancerrxgene.org platform). Interestingly, we observed different top-scoring kinases compared to patients (Appendix A). Top-scoring kinase coding genes for leukemia cell lines were FLT3, NTRK1, INSR, RYK, FGFR1, MERTK, ROR2, FGFR3, KIT, RET, ROR1, PTK7, DDR2, DDR1, and CSF1R; NTRK1, PDGFRA, ALK, AXL, RET, AATK, ROR2, TYRO3, INSR, DDR1, NTRK3, DDR2, FGFR3, FGFR1, and FLT3—for NB cell lines (Figure 2b). FLT3, NTRK1, INSR, FGFR1, ROR2, FGFR3, RET, DDR2, and DDR1—common for the cell lines of both origins. More importantly, only NTRK1 and FGFR3 were detected to be high-scoring among both leukemia and NB patients and cell lines (Appendix A). The observed difference in top-scoring kinases within cell lines and patient samples could be explained by the infiltration of cancer patient samples with normal (non-cancerous) cells including immune cells and the additional regulation of gene expression by extracellular molecules.

To further characterize the differences of RTK-coding genes in NB and leukemia cells, we plotted the HDAC SCORE of each RTK-coding gene against its average expression in cells (Figure 2c). The distinctive HDAC SCORE and expression were specific for FLT3 (leukemia) and NTRK1 (NB).

### 3.3. Hydrazostat Induce the Expression of RTK-Coding Genes in NB and Leukemia Cells

To investigate whether the expression of RTK depends on HDAC class I and II activity, leukemia cells Kasumi-1, K562, THP-1, HL-60, and NB cells LAN-1, SH-SY5Y, and Kelly were incubated with cytotoxic concentrations (according to IC50 value for hydrazostat for each cell line) of belinostat and hydrazostat for 72 h. We measured the expression of genes coding RTKs: *FLT3*, *KIT* exclusively in leukemia cell lines, *PDGFRa*, *AXL* in NB, *NTRK1*, *FGFR3*, *INSR*, *ROR2*, and *RET* in both NB and leukemia cells by qRT-PCR, which were top-scoring RTK-coding genes among leukemia and NB cell lines (Figure 2a,b).

*KIT* mRNA level was upregulated (3.5-fold) significantly in HL-60 and K562 cells exposed to hydrazostat, but not to belinostat (Figure 3a). In contrast, we detected slight downregulation of *KIT* expression in Kasumi-1 and THP1 cells incubated with hydrazostat (Figure 3a). We detected upregulation (10-fold) of *FLT3* in THP-1 cells and downregulation of this gene in HL-60 cells treated with hydrazostat. At the same time, *FLT3* expression was repressed in response to belinostat in HL-60 cells.

Dramatic induction (433-fold) of *AXL* mRNA was a unique feature of hydrazostat-treated SH-SY5Y cells. Notably, *AXL* mRNA was not increased in LAN-1 and Kelly cells. What is important, the effect was not dependent on basal *AXL* expression in studied cells, as LAN-1 cells have relatively low *AXL* expression levels and SH-SY5Y and Kelly-high (Appendix A). *PDGFRa* expression remained unchanged in the majority of cases except for belinostat-treated SH-SY5Y cells (Figure 3a).

We observed *NTRK1* induction by belinostat and hydrazostat in the majority of studied cells (HL-60, Kasumi-1, THP-1, LAN-1, SH-SY5Y, and Kelly). Notably, the effect was more prominent in hydrazostat-treated cells and the fold change was superior in leukemia (150-fold for Kasumi-1) compared to NB cells (8-fold for Kelly cells) (Figure 3c and Appendix A). Additionally, a significant increase in *ROR2* was detected in hydrazostat-treated THP-1 and Kelly cells (Figure 3c and Appendix A). *ROR2* mRNA was not detected in Kasumi-1 cells treated with belinostat. No modulation of *INSR* was detected among tested cell lines.

As we detected a profound upregulation of FLT3 mRNA in THP-1 (>10x) and KIT mRNA (>3x) in K562 cells cells in response to hydrazostat, we decided to study FLT3 and KIT proteins in these cells. For that purpose, we stained THP-1 cells pre-incubated with 2 μM belinostat, hydrazostat or DMSO for 72 h with anti-FLT3 antibodies. At the same time K562 cells were stained with anti-KIT antibodies after incubation with 1 μM belinostat, hydrazostat or DMSO for 72 h. We revealed notable accumulation of FLT3, a protein in the cytoplasm, in hydrazostat- but not belinostat-treated THP-1 cells (Figure 3d). A similar effect was observed in K562 cells. Thus, we confirmed the ability of hydrazostat to induce mRNA and protein of *FLT3* and *KIT* genes.

### 3.4. Hydrazostat Makes Cells More Sensitive to RTK Inhibitor Imatinib and Cytotoxic Drugs

Based on our findings that hydrazostat induces the expression of RTK-coding genes we evaluated the activity of hydrazostat in combination with RTK inhibitor. We exposed four leukemia and four NB cells to multi-kinase (KIT, PDGFRa, and PDGFRb) inhibitor-imatinib [44] in the presence of belinostat or hydrazostat. We selected imatinib for our analysis as it shows promising but limited efficacy in clinical trials for NB and AML (NCT00030667 and NCT01126814) and is successfully used for chronic myeloid leukemia treatment, and both types of cancer express *KIT*, which is one of the main imatinib target. Additionally, KIT was one of the top-scoring genes with the highest HDAC SCORE for leukemia patients and was among top genes in NB patient analysis. Additionally, we have studied data on the sensitivity of AML, CML and NB cells to several FGFR-family (AZD4547, FGFR_3831, and PD173074), NTRK-family (AZD1332 and GW441756) and KIT inhibitors (imatinib, sunitinib, and sorafenib) available at CancerRxGene [45] and revealed that average geometric mean of KIT inhibitors was lower than for FGFR and NTRK targeting drugs (Appendix A).

Overall, for 100% (4/4) leukemia and 100% NB cell lines, we detected synergy of hydrazostat in combination with imatinib compared to 50% (2/4) and 66.7% (2/3) for belinostat (Figure 4 and Appendix A). Synergy scores for the combination of hydrazostat with imatinib were significantly higher compared to belinostat (Appendix A).

Notably, the most prominent difference between the Bliss synergy scores of imatinib combination with belinostat or with hydrazostat was detected in Kasumi-1 (1.2 and 17.96) and SH-SY5Y (−1.5 and 16.5) cells (Figure 4). Kasumi-1 cell line bears *KIT* mutation (*N822K*) which makes this kinase ligand-independent and confers an unfavorable prognosis in CBF-AML. According to recently published work, blockade of mutant RTKs in combination with HDAC class I inhibitors shows a more pronounced toxic effect [46]. Interestingly, belinostat/imatinib combination possessed a superior Bliss synergy score than hydrazostat in Kelly cells despite the increased sensitivity of these cells to hydrazostat, which means that observed synergy is independent of HDAC inhibitors toxicity but relies on molecular mechanism of the studied compounds. 

To investigate whether hydrazostat can enhance the cytotoxic effect of clinically approved chemotherapy drugs, we treated leukemia and NB cells with cytarabine and vincristine, respectively. For the majority of tested cell lines (6/7), the combination of hydrazostat with cytotoxic drugs possesses a profound synergistic effect compared to belinostat, indicating that combined treatment is active across a broad spectrum of leukemia and NB subtypes. As the exception, no synergy between hydrazostat and cytarabine was observed in Kasumi-1 cells. Again, as for imatinib, synergy scores of hydrazostat with cytarabine or vincristine were significantly higher than for belinostat (Appendix A).

To describe the mechanism of the increased cytotoxicity of imatinib in combination with hydrazostat in more detail, we have studied the activity of ERK kinase—a major component of RTK signaling pathways—as we have recently shown the dependence of neuroblastoma cell survival under imatinib on ERK activity [39]. For that purpose, we have used SH-SY5Y with ERK-KTR reporter and measured ERk activity in live individual cells incubated for 24 h with DMSO, 30 μM imatinib, 5 μM belinostat, 5 μM hydrazostat or their combinations. We show that hydrazostat and imatinib significantly (*p*-value < 0.001) induced ERK activity in SH-SY5Y cells (Appendix A). Despite that belinostat alone have not influenced ERK activity, its combination with imatinib enhanced it (*p*-value = 0.0019). ERK activity in SH-SY5Y cells treated with hydrazostat and imatinib was lower than in belinostat and imatinib treated cells and comparable to hydrazostat and imatinib added as single agents. We conclude that hydrazostat enhances ERK activity—major component of RTK signaling pathway, along with the elevation of RTK-coding genes at the mRNA level, but the addition of imatinib is not impactful, thus causing cell death but not survival. We have also studied phosphorylated ERK in K562 leukemia cells with confocal microscopy. We show that hydrazostat and belinostat does not drive dramatic upregulation of ERK activity, but both hydrazostat and belinostat lead to the downregulation of ERK when combined with imatinib (Appendix A).

Thus, we have demonstrated that hydrazostat could enhance the toxicity of kinase inhibitor imatinib and cytotoxic drugs—cytarabine and vincristine—that are widely used for NB and leukemia treatment; and the high synergy scores of the drug combinations are accompanied with ERK activity downregulation in the cells.

### 3.5. Hydrazostat Enhances Venetoclax-Induced Apoptosis

We suggested that the observed difference in toxicity of hydrazostat and belinostat could be realized through the apoptosis. Hydrazostat treatment led to the induction of apoptosis in up to 18% of cells (Figure 5a).

Additionally, we examined the effect of belinostat and hydrazostat on the expression of *BCL2*, a negative regulator of apoptosis and the target of anticancer drug venetoclax, in THP-1 and Kelly cells. *BCL2* was upregulated by hydrazostat but not belinostat in THP-1 cells (Appendix A).

We reasoned that BCL2 inhibitor venetoclax could enhance the toxicity of hydrazostat. We detected a significant accumulation (up to 40%) of Annexin V-positive in both THP-1 and Kelly cells in response to hydrazostat and venetoclax combination (Figure 5a,b). Co-exposure of AML and NB cells to hydrazostat and venetoclax led to a profound reduction in cell number in all tested cell lines and resulted in high synergy scores for the combination of these drugs (Appendix A). We were not able to proceed with apoptosis detection in SH-SHSY cells, because the reduction in cell number in response to hydrazostat and venetoclax combination was dramatic.

## 4. Discussion

Several potent and selective HDAC inhibitors with limited off-target effects have been developed [47,48] and they serve as promising powerful tools to reverse pathological imbalance in acetylation/deacetylation of histones within various diseases. It is known that HDACs have tissue-specific transcription activity in normal and malignant cells. Despite the broad implication of multitargeting HDAC inhibitors in clinical practice (romidepsin, belinostat, panobinostat, and vorinostat) and trials (abexinostat, quisinostat, and resminostat), little is known about the relevance of individual HDACs as potential targets in leukemia and NB patients. As HDAC inhibitors could return transcriptomic signature induced by chemotherapeutic drugs such as cytarabine to normal state and target leukemia stem cells population of cancer cells [49], they could have promising activity in combination with known anticancer drugs. What is more important, the efficacy of single-target agents is often limited in their clinical utility because tumors harbor multiple dysregulated growth and survival pathways, which evolve during the course of treatment.

Within this work, we aimed not only at the investigation of the HDAC class I novel inhibitor hydrazostat’s anticancer effectiveness in leukemia and neuroblastoma cell models, but also at the identification of the additional vulnerabilities of cancer cells tightly linked with HDACs. As HDACs act as co-regulators of oncogenes expression and RTK are among most studied cancer drivers, we decided to investigate the connection between these two classes of proteins in leukemia and NB cells. The first part of this study was devoted to the identification of RTK regulated by individual HDACs and, especially, HDAC class I. Computational analysis of RNA-sequencing data of 613 leukemia and NB patient samples and 88 cancer cell lines of the hematopoietic and neural origin, allowed us to identify *NTRK1*, *FGFR3*, *AXL*, *KIT*, *PDGFR, ROR2*, *FLT3*, *RET*, and *INSR* association with HDACs. We tested the ability of hydrazostat and belinostat to affect the expression of RTK revealed within the study. What has driven our attention is the ability of hydrazostat to reactivate *NTRK1* in studied cancer cells, especially in AML t(8;21) cells-Kasumi-1. Selective anti-HDAC8 inhibitors are known to induce differentiation of NB cells with concomitant induction of *NTRK1* [32,50], and HDAC1 in complex with MYCN suppresses NTRK1 expression [51]. Additionally, evidence of upregulation of NGF and phospho-TrkA after treatment with HDAC inhibitors was detected in astrocytes [52]. Although high *NTRK1* expression is a favorable prognostic marker in NB and a marker of differentiation, its role in AML cells remains unclear. Previously, we have reported the similarities in TrkA regulation in leukemia and NB cells [1], and our recent findings expand this study. What is important, we observed dramatic induction of AXL expression in SH-SY5Y cells by HDAC class I inhibitor, which could be beneficial for the simultaneous repression of AXL and HDAC, as it was described for glioma [53]. It was shown that acquired resistance of SH-SY5Y cells to ALK inhibitors was associated with the induced AXL expression, but AXL downregulation made the cells more sensitive to ALK inhibition [54]. Of note, the direct mechanism of transcription regulation of *AXL* was not shown, and thus our findings provide novel insights into the possible mechanism of AXL regulation by HDAC proteins, which could be studied further.

Next, we investigated the ability of RTK inhibitor imatinib to enhance the potency of HDAC inhibitors. The efficacy of HDAC inhibitors in combination with tyrosine kinase inhibitor imatinib was previously described for CML cells [55]; however, pan-HDAC inhibitors often exhibit high toxicity in patients. Exclusive sensitivity of FLT3 mutant cells to dual inhibition of HDAC and RTK was shown previously [56]; however, our data demonstrate the existence of a more complex mechanism, suggesting that not only genetic mutation drive the vulnerability but also the mRNA level of RTK-coding genes. Although the enhancement of the cytotoxicity of FLT3 inhibitors with HDAC inhibitor dacinostat was registered for MV-4-11 cells and primary leukemia cells, this effect was specific for AML cells harboring FLT3 activating mutation [57]. All in all, in our research, we have not focused on mutant RTKs and thus expanded the therapeutic potential of HDAC inhibitors on leukemia and NB cells with different genomic and transcriptomic features. Mutant RTK are detected only in limited number of patients, thus defining the need of the investigation of non-mutant RTKs in cancer cells. We revealed the implication of HDAC in the regulation of RTKs and suggest use it as an advantage for leukemia and neuroblastoma treatment with RTK and HDAC inhibitors. Surprisingly, while studying the ability of hydrazostat to induce apoptosis in leukemia and NB cells, we observed the differential regulation in *BCL2* expression. As the synergy between HDAC inhibitors and venetoclax (BCL2 inhibitor) was shown for patients with advanced-stage cutaneous T-cell lymphoma and glioblastoma [58,59], we researched this phenomenon in leukemia and NB sells. We found that hydrazostat enhances venetoclax-induced apoptosis in these cells in a synergetic manner.

Our findings reveal that reactivation of RTK expression in response to HDAC inhibition could be used for combinational therapy, which can be a useful strategy to overcome the complications of single-agent therapies.

## 5. Conclusions

Hydrazostat—a novel selective HDAC class I inhibitor—has superable toxicity against most NB and leukemia cells.

RTK-coding genes differentially depend on HDAC expression in patient samples and cancer cell lines.

*NTRK1* and *FGFR3* are HDAC class I-associated genes in leukemia and NB patients and cell lines.

HDACs class I inhibition with hydrazostat reactivates *NTRK1*, *FLT3*, *KIT*, *ROR2*, and *FGFR3* expression in at least 30% of studied cell lines.

Hydrazostat enhances the sensitivity of NB and leukemia cells to RTK inhibitor imatinib, cytotoxic drugs cytarabine and vincristine and BCL2 inhibitor venetoclax. Thus, hydrazostat could be used to overcome the complications of single-agent therapies.

## Figures and Tables

**Figure 1 biomedicines-09-01846-f001:**
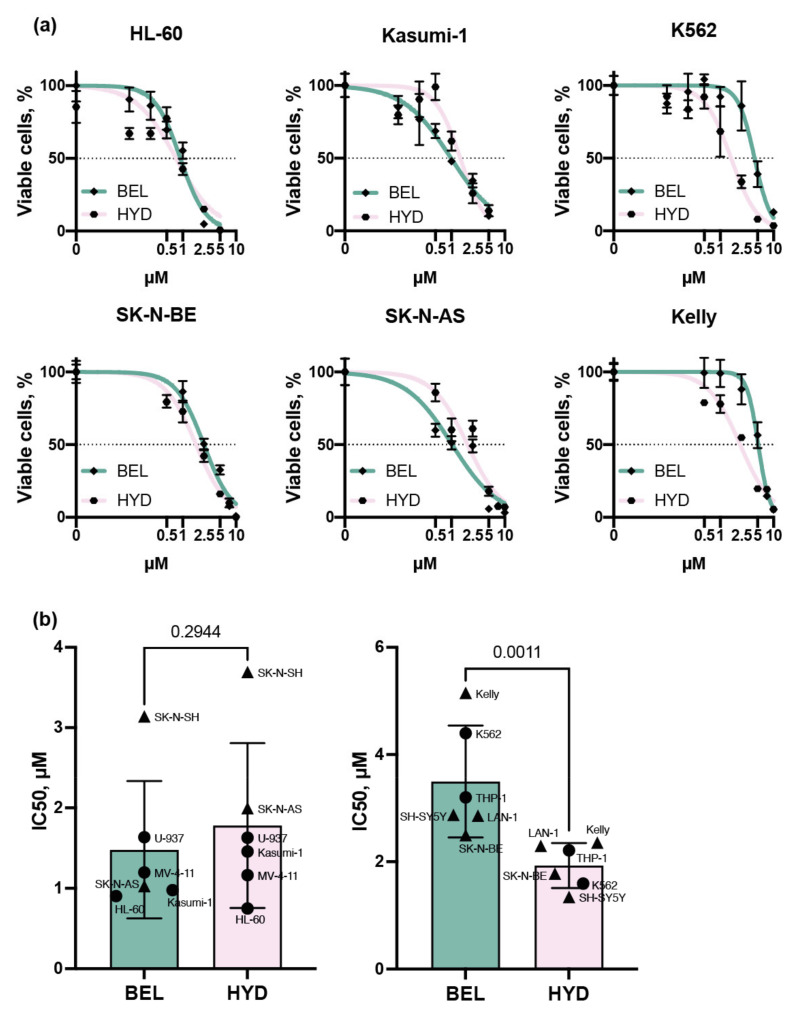
The sensitivity of leukemia and NB cells to belinostat (BEL) and hydrazostat (HYD). (**a**) Viability of leukemia (HL-60, Kasumi-1, and K562) and NB (LAN-1, SK-N-AS, and Kelly) cells treated with belinostat (green) and hydrazostat (pink) relative to DMSO treated cells. (**b**) IC50 values of belinostat (green) and hydrazostat (pink) of leukemia (circle) and NB (triangle) cell lines with comparable sensitivity of belinostat and hydrazostat (left) and increased sensitivity to hydrazostat (right). All experiments were performed in three replicates. Cells were counted by trypan blue exclusion on day 3 after the addition of inhibitors. *p*-value was determined by Mann–Whitney test.

**Figure 2 biomedicines-09-01846-f002:**
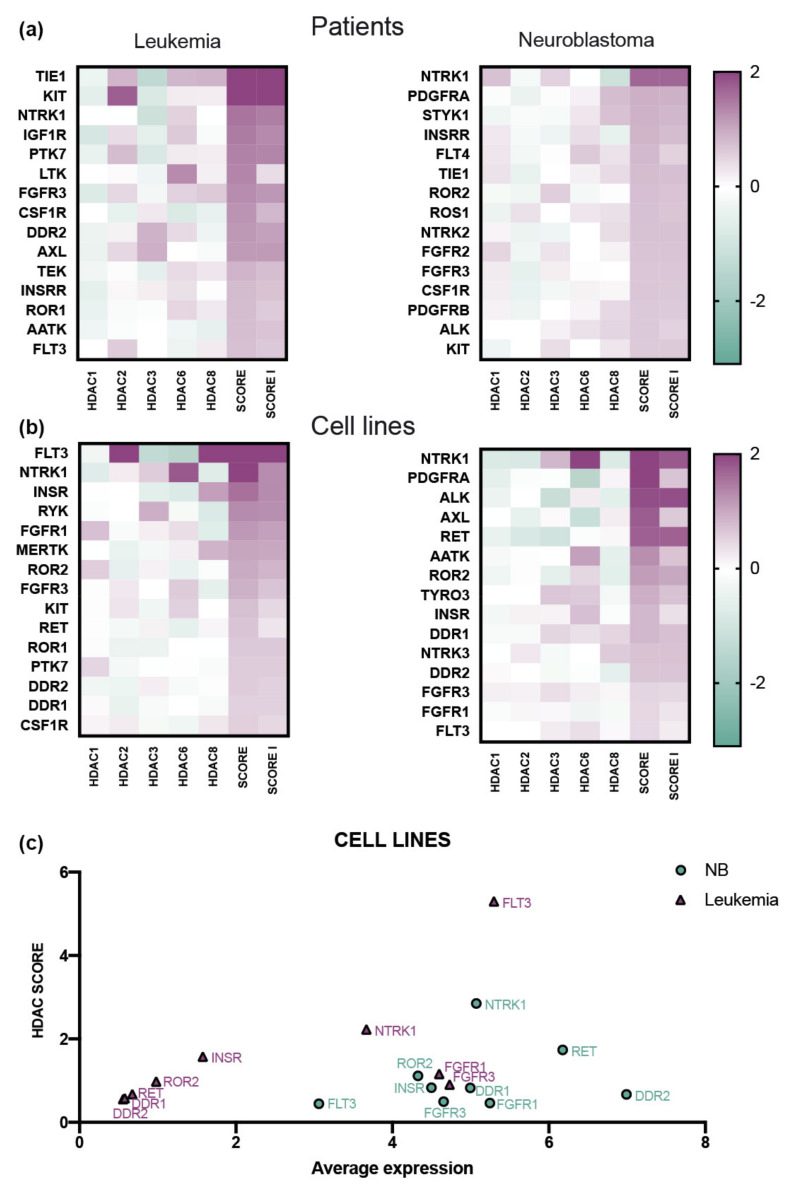
Correlation of HDACs genes expression with all human receptor tyrosine kinases. (**a**) Elastic net coefficients of HDAC class I and II with RTK-coding genes in AML (*N* = 525) and NB (*N* = 88) patients. The color indicates the value of the correlation coefficient determined by the elastic regression method. Genes are ranged by the SCORE that is determined as the sum of squared coefficients for HDAC1, 2, 3, 6,8, SCORE I is determines as the sum of squared coefficients for HDAC1, 2, 3, 8. Top 15 genes according to the SCORE are shown. (**b**) Correlation coefficients of HDAC class I and II with RTK-coding genes in leukemia and NB cell lines. The color indicates the value of the correlation coefficient determined by the elastic net regression model. Genes are ranged by the SCORE that is determined as the sum of squared coefficients for HDAC 1, 2, 3, 6, 8, SCORE I is determined as the sum of squared coefficients for HDAC 1, 2, 3, 8. Top 15 genes according to the SCORE are shown. (**c**) Average expression of gene vs. its SCORE in cell lines. Average expression was determined for 59 leukemia and 29 neuroblastoma cell lines. NB are presented as green circles, and leukemia as purple triangles.

**Figure 3 biomedicines-09-01846-f003:**
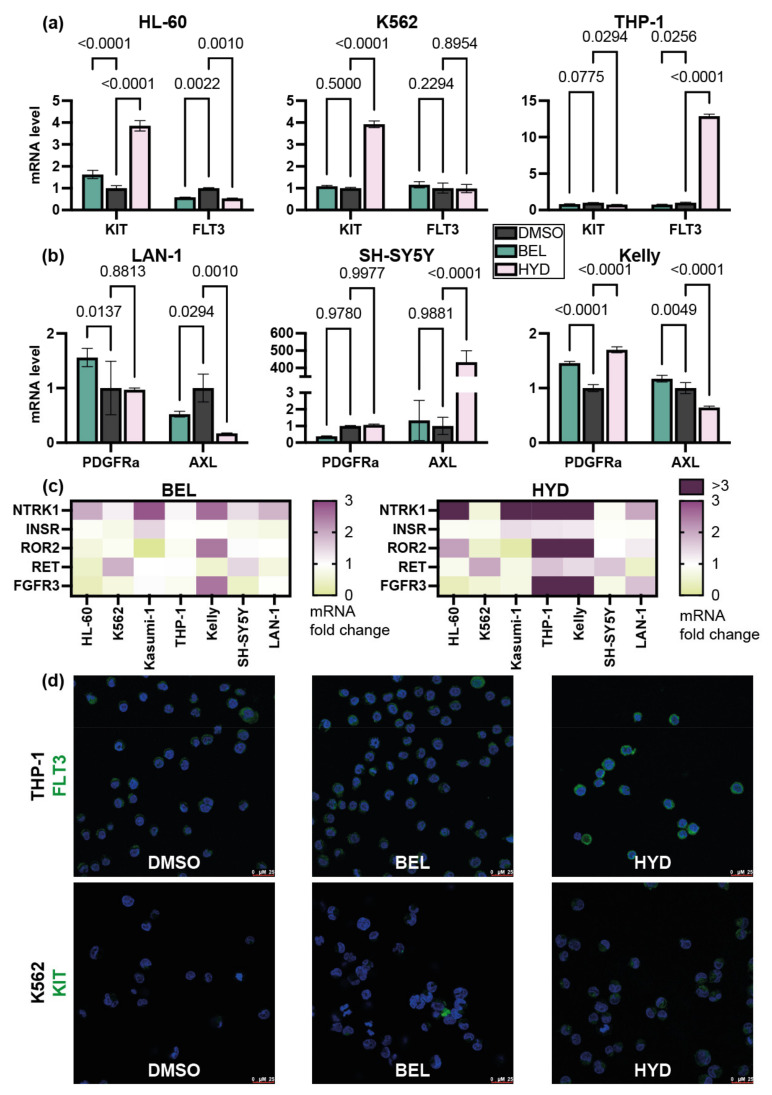
Changes in mRNA level of receptor tyrosine kinases in response to belinostat (BEL) and hydrazostat (HYD) measured by qRT-PCR. (**a**) *KIT* and *FLT3* mRNA levels in leukemia cells incubated with DMSO, belinostat, or hydrazostat for 72 h. (**b**) *PDGFRa* and *AXL* mRNA levels in NB cells incubated with DMSO, belinostat, or hydrazostat for 72 h. (**c**) *NTRK1*, *INSR*, *ROR2*, and *RET* mRNA in leukemia HL-60, K562, Kasumi-1, THP-1, and NB Kelly, SH-SY5Y, LAN-1 cells incubated with belinostat or hydrazostat for 72 h. The color indicates a fold change of gene expression relative to control (DMSO). Fold changes higher than 3 are represented as bars in Appendix A. (**d**) THP-1 cells were pretreated with DMSO, belinostat, or hydrazostat for 72 h and stained with an anti-FLT3 antibody (green) and DAPI dye (blue). K562 cells were pretreated with DMSO, belinostat, or hydrazostat for 72 h and stained with an anti-KIT antibody (green) and DAPI dye (blue). qRT-PCR was performed in three replicates. P-value was calculated by two-way ANOVA with no correction.

**Figure 4 biomedicines-09-01846-f004:**
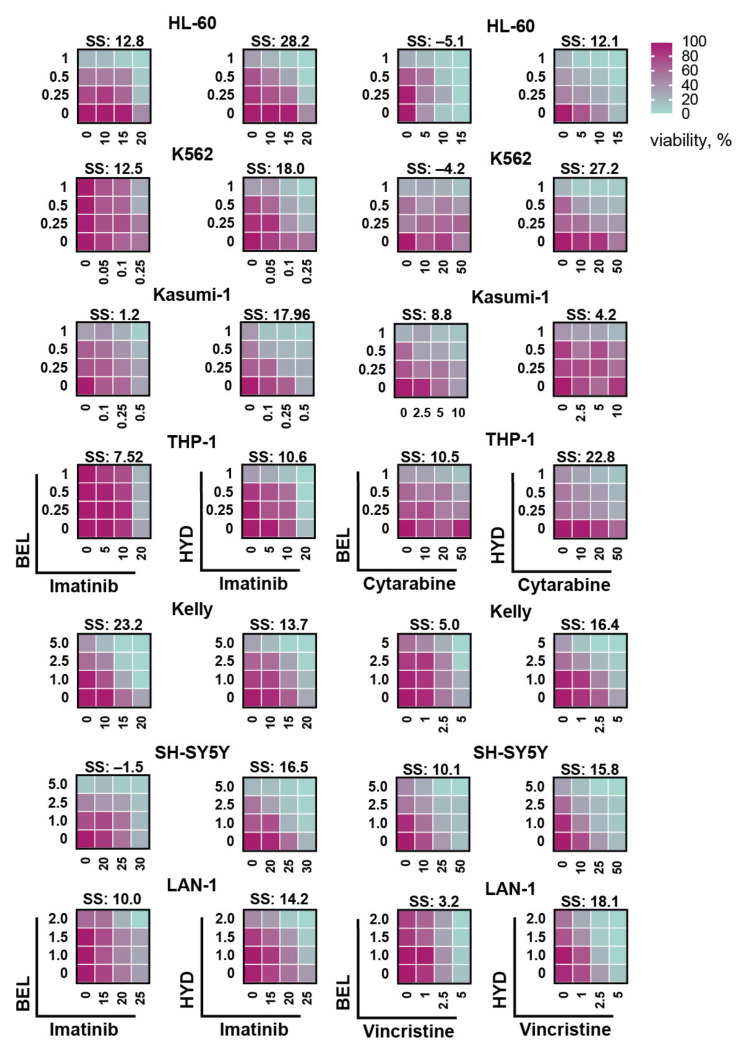
Viability of cells co-treated with imatinib and belinostat (BEL) or hydrazostat (HYD), cytarabine (leukemia), or vincristine (NB) with belinostat or hydrazostat. Bliss synergy scores (SS) drug combinations were calculated with SynergyFinder. The color corresponds to % of viable cells ranging between 0 (green) and 100 (red). The original 2D plots are presented in Appendix A. Belinostat, hydrazostat, and imatinib concentrations are represented in uM, cytarabine and vincristine in nM. Drugs were applied simultaneously directly to the cell culture media and cells were incubated with drugs for 6 days before the count. All experiments were performed in three replicates. Bliss synergy score exceeding 10 corresponded to a synergistic effect of two compounds.

**Figure 5 biomedicines-09-01846-f005:**
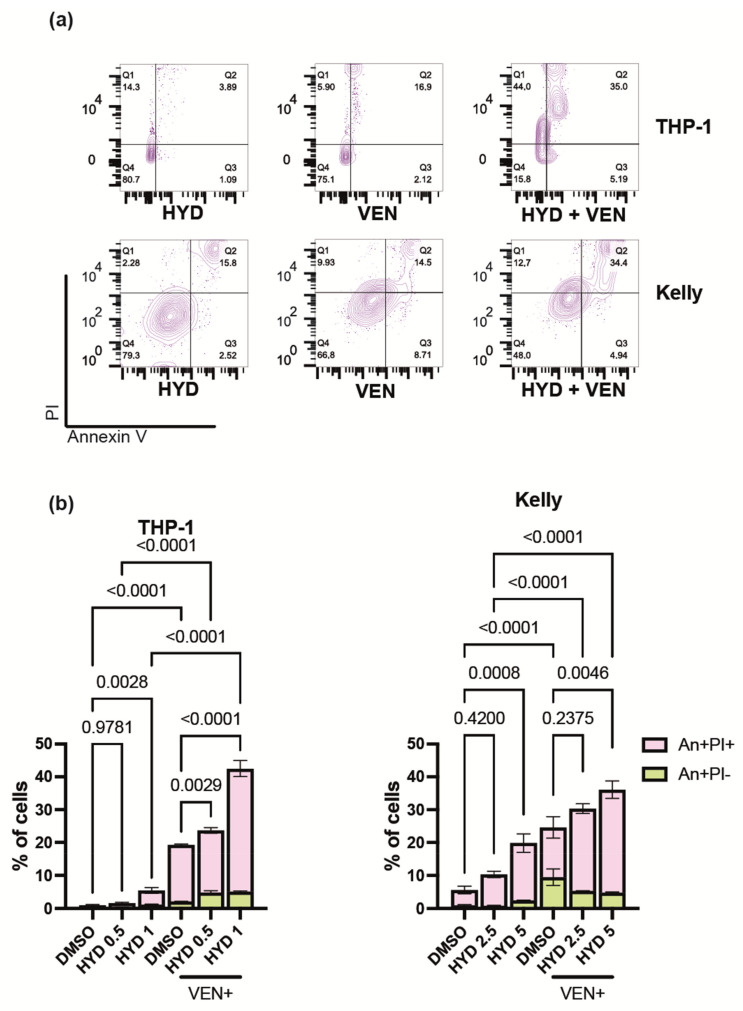
Hydrazostat enhances venetoclax-induced apoptosis. (**a**) Dot plots acquired by flow cytometry showing the percent of Annexin V and PI positive THP-1 and Kelly cells incubated with venetoclax, hydrazostat or their combination for 6 days. The hydrazostat concentration used for THP-1 is 1 μM and 5 μM for Kelly; 100 nM of venetoclax was used for THP-1 cells and 5 μM for Kelly. (**b**) Percent of apoptotic cells detected by flow cytometry analysis of cells stained with Annexin V-FITC and Propidium Iodide. Hydrazostat concentration is indicated in μM; 100 nM of venetoclax was used for THP-1 cells and 5 μM for Kelly. This experiment was performed in three replicates; SEM is shown for each bar. *p*-value was determined by one-way ANOVA with no correction.

## Data Availability

Code is available at CancerCellBiology, HitHub [38].

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
