# Peer review of "Selective Inhibition of HDAC Class I Sensitizes Leukemia and Neuroblastoma Cells to Anticancer Drugs"

_biomedicines, 2021, doi:10.3390/biomedicines9121846_

Round 1

Reviewer 1 Report

The manuscript “Selective Inhibition of HDAC Class I Sensitizes Leukemia and Neuroblastoma Cells to Anticancer Drugs” described a useful strategy to overcome the complications of single-agent therapies in the leukemia and neuroblastoma cells. The authors developed novel belinostat derivative named hydrazostat which targets HDAC class I and revealed the connection between HDAC class I and receptor tyrosine kinase (RTK). They also found that the ability of hydrazostat to enhance venetoclax-induced apoptosis. This work is meaningful and pertinent, and the conclusions are mainly supported by the corresponding results to some extent. The referee agrees to recommend this manuscript for the publication after the following revision.

  1. Your manuscript needs to be carefully edited and polished in terms of grammar and sentence structure before resubmission.
  2. The caption of the figure legends such as the Figure 2b and the Figure 2c should be revised to a unified style, either phrases or sentences.
  3. There are some small mistakes in your manuscript layout that you need to check again. For example: in section 3.1, paragraphs 4 and 5.
  4. In Figure 3a, Figure 3b and so on, the P-values need to be added to the displayed images. If there is no difference in P values, the symbol is “ns”.
  5. In this work, the hydrazostat could be used to overcome the complications of single-agent therapies and it has superable toxicity against most NB and leukemia cells. For further clinical significance, you should continue to conduct subsequent animal studies.

Author Response

We would like to thank the Reviewer for the thoughtful comments. We believe that we have improved our manuscript according to the Reviewer’s suggestions.

  1. We have carefully revised the manuscript and corrected grammar mistakes and changed the structure of some sentences. All changes are highlighted in blue.
  2. The captions of the figures have been unified.
  3. We have carefully checked the layout of the manuscript and made the corrections.
  4. P-values corresponding to NS p-value (>0.05) were previously omitted, but in the current version of the manuscript we have added numerical p-values to all graphics according to the results of the statistical test that was performed.
  5. We understand that to achieve more clinical significance the ability of hydrazostat and its combination with imatinib to inhibit the growth of tumors in animal models should be tested, but we hope that based on out in vitro results and bioinformatical analysis, in vivo studies will be performed in the future. At this point, we reveal novel important mechanism of the reactivation of RTK genes expression in response to HDAC inhibitors and show its usefulness for the development of the effective combinational therapy for leukemia and neuroblastoma patients.

Reviewer 2 Report

Review of Biomedicines:1439021 titled “Selective Inhibition of HDAC Class I Sensitizes Leukemia and Neuroblastoma Cells to Anticancer Drugs”

In this study, Vagapova et al. investigate the therapeutic potential of the HDAC inhibitor hydrazostat in leukemia and neuroblastoma. This study is important because it demonstrates the effectiveness of the two-punch strategy for cancer treatment where the first inhibitor sensitizes cancers to the second inhibitor. Normally activation of kinases is a major mechanism for drug resistance, but it also presents itself as an opportunity for combination treatments. While the authors should be commended for their approach of using publicly available patient data along with experimental data, the major findings of the manuscript need to be strengthened before reconsideration for publication in the special issue Special Issue "The Contribution of Epigenetics to Normal and Malignant Hematopoiesis".

Major comments:

  1. The authors should demonstrate changes in the kinase protein expression like they did in Figure 3d for KIT in HL-60 and K562 cell lines, as KIT is an important kinase in this study regarding the effect of Imatinib + HYD combination.
  2. Changes in mRNA or protein expression of kinases do not always translate into changes in activity. The authors should demonstrate that increased KIT, FLT3, AXL, or NTRK1 expression upon HYD treatment corresponds to increased activity by looking at phosphorylation of downstream kinases via Western Blots.
  3. While their combination might be effective against cancer cells, it might also be toxic to normal cells. The authors should test the toxicity of their drug combination(s) on normal/healthy cell lines

Minor comments:

  1. The authors’ rationale for combining HYD with KIT inhibitor imatinib is reasonable (though there is a significant concern about non-specificity of this inhibitor for the purposes of this study as it also inhibits ABL and PDGFR kinases very potently). However, it seems like all of their data consistently point towards NTRK1 as the major player. NTRK1 data presented in the manuscript seem consistent between patients and cell lines, and the gene has a high score in their elastic net predictions. More importantly, there are multiple NTRK inhibitors that can be tested in combination with HYD (PMID: 33328556). The authors should explain why they decided not to follow NTRK1 and speculate what effect NTRK inhibitor – HYD combo might have on the cancer types tested. Similar commentary can be made about FGFR3 as well.
  2. Line 15 - the sentence is grammatically incorrect
  3. Line 169:170 – the sentence does not make sense
  4. Line 194 – “squired”…
  5. Figure 2 legend - be consistent with “SCORE” vs “score” vs “score” throughout the manuscript
  6. Figure 2c - what do the four large circles indicate? Different quadrants? Clusters?
  7. Line 66 – “chimeric protein” do you mean fusion proteins?
  8. Figure 1b – indicate what triangle vs. circle mean in the legend
  9. Lines 216-219 – Or it is because of the limitations of cell lines in recapitulating human disease. The cell lines do change over time because they get used to growing on petri dishes and they accumulate further mutations.
  10. Lines 231 – “Figure 2a and a” ?
  11. The authors should briefly mention how many HDACs there are and how different/similar they are in the Introduction to make the manuscript more accessible to out-of-field readers.

Author Response

First, we would like to thank the reviewer for the valuable comments. We believe that with the additional experiments performed as suggested by the Reviewer 2 we have sufficiently improved the quality of the manuscript and strengthened the conclusions.

Response to the Major comments of Reviewer 2:

  1. We have treated K562 cells with 1 uM hydrazostat, belinostat or DMSO for 72 hours, and stained cells with anti-KIT antibodies conjugated with FITC (ab119107, Abcam, USA). We have selected K562 cells to study KIT at protein level because a) previously we have compared KIT protein levels in HL-60 and K562 cells (by flow cytometry and confocal microscopy) and demonstrated almost the same level in these cells [1] b) the highest synergy score for hydrazostat and imatinib was observed for K562 cells cell. We show that hydrazostat drives accumulation of KIT protein in the cytoplasm of the cells upon treatment compared to DMSO-treated cells. At the same time, belinostat did not cause any significant changes. Also, we show that KIT is distributed all over cytoplasm in response to hydrazostat. Notably, in K562 cells incubated with DMSO or belinostat only a part of the population was KIT-positive, however, we have detected KIT protein in the cytoplasm of the majority of cells treated with hydrazostat.
  2. We fully agree that changes of mRNA and protein levels in cells often do not correspond the activity of studied kinases and signaling pathways. As suggested by the reviewer we have studied activity of one of the KIT/FLT3 downstream kinases – ERK. ERK kinase is the crucial component of the most RTK signaling pathways and ERK activity is tightly linked with cell survival. Recently [2] we have shown that ERK activity in neuroblastoma cells is an important marker of cell survival and ERK inhibition leads to cell death. We used SH-SY5Y:ERK-KTR cell line established as described in [2] to measure ERK activity in live neuroblastoma cells. Briefly, ERK-KTR translocation reporter was introduced in SH-SY5Y cells by the lentiviral transduction. For that purpose, we used pLentiCMV Puro DEST ERKKTRClover plasmid (Addgene plasmid #59150), which encoded ERK docking domain ELK, nuclei localization site and nuclei extraction site (containing phosphorylation sites) fused with fluorescent protein mClover. SH-SY5Y:ERK-KTR cells were incubated with DMSO, 5 uM belinostat, 5 uM hydrazostat, 30 uM imatinib or their combination for 24h before the analysis. Cells were imaged with Leica DMI8 automated microscope using 10x magnification lenses. Nuclei of the cells were stained with 500 ng/ml Hoechst-33342 for 30 min before imaging. For each drug/combination we used three wells and three fields were imaged in each well. Imaging of the cells was performed in two channels – 488 nM (for Hoechst) and 520 nM (for mClover). Illumination correction, segmentation of nuclei and cells, calculation of cytoplasm to nucleus ratios of individual cells (C/N ratio) corresponding to ERK activity were made in CellProfiler. We show that hydrazostat (p-value = 0.0009) and imatinib (p-value = 0.0009) significantly induced ERK activity in SH-SY5Y cells (Figure S4d). Despite that belinostat have not made any changes to ERK activity, but its combination with imatinib enhanced ERK activity (p-value = 0.0019). ERK activity in SH-SY5Y cells treated with hydrazostat and imatinib was lower than in belinostat and imatinib treated cells and comparable to hydrazostat and imatinib added as single agents. We conclude that hydrazostat enhances the activity of ERK kinase – major component of RTK signaling pathway, but imatinib does not enhance this activity, thus not driving cell survival. Additionally, we have stained K562 cells treated with belinostat, hydrazostat, imatinib or their combinations with anti-phospho-ERK1/2 antibodies. We show that both hydrazostat and belinostat lead to the downregulation of ERK in combination with imatinib (Figure S4e).
  3. We agree that the assessment of the drug toxicity to normal cells has great importance for the clinical significance, however, according to the published papers 500 nM of belinostat cause only 1% death of lymphocytes (PBMC) [4] and the concentrations of both HDAC inhibitors used by us in this study varied from 250 to 5000 nM. Also, the most synergistic area of hydrazostat’s combination with imatinib and cytarabine/vincristine starts from very low concentrations of HDAC inhibitor - 250 nM (Figure S5).

Response to the Minor comments:

  1. We have extended the 3.4 section and provided brief information on the reason of choosing imatinib for combinational studies. Indeed, we have found that NTRK1 and FGFR3 have high correlation coefficients (Elastic NET coefficients) with HDAC coding genes in both patient cells and cell lines, however, basal NTRK1 expression in leukemia cells is rather low – most of the cells has predominant TrkAIII mRNA isoform is dominating and protein is localized in spike-like compartments near the nuclei but not in the cytoplasm or cell surface [1]. Second, we have studied IC50 values of inhibitors of several kinases of FGFR-, NTRK- families determined for neuroblastoma and leukemia cells available at https://www.cancerrxgene.org. We have found three FGFR-family inhibitors (AZD4547, FGFR_3831, PD173074) and two NTRK-family inhibitors (AZD1332, GW441756) and three KIT inhibitors (Imatinib, Sunitinib, Sorafenib). As an example, THP-1 cells had the lowest sensitivity to FGFR_3831 (FGFR1, FGFR2, FGFR3, FGFR4 inhibitor) – 72.1 uM and the highest to imatinib - 13.8 uM. Kelly cells were resistant to AZD1331 (NTRK1, NTRK2, NTRK3 inhibitor) - 38.3 uM and sensitive to AZD4547 (FGFR1, FGFR2, FGFR3 inhibitor). K562 cells had the smallest IC50 value for imatinib 0.4 uM and the largest for PD173074 (FGFR1, FGFR2, FGFR3 inhibitor) 59.6 uM. All in all, geometric means of IC50 values of FGFR, NTRK, KIT inhibitors calculated for AML, CML and NB cell lines varied from 0.5 uM (Imatinib, CML) to 35.1 uM (PD173074, NB). All in all, average IC50 value of KIT inhibitors which also target PDGFR and ABL1 for AML, CML and NB cells was the lowest (Figure S4a).

Table 1. Geometric mean of IC50 values (uM) of FGFR, NTRK, KIT inhibitors calculated for AML, CML and NB cell lines according to https://www.cancerrxgene.org.

AML

CML

NB

AZD4547

5.8

4.3

13.4

FGFR_3831 

7.3

9.3

13.0

PD173074

16.2

19.6

35.1

AZD1332

14.3

14.8

26.2

GW441756

11.3

14.7

9.9

Imatinib

13.6

0.5

18.2

Sunitinib

2.63

6.2

21.3

Sorafenib

4.0

3.3

7.7

What is also important the most TrkA inhibitors such as Larotrectinib (described in PMID: 33328556) are effective against TrkA-fusion proteins such as TPM3-NTRK1 and ETV6-NTRK3 and its activity against wild-type TrkA protein is not described. In our experiments we show that 20 uM Larotrectinib cause minimal toxicity to the majority of AML cell lines.

2. Line 15 – we have corrected the sentence

3. Line 169:170 – we have corrected this sentence

4. Line 194 – “squired”… - we have corrected this mistake

5. Figure 2 legend - be consistent with “SCORE” vs “score” vs “score” throughout the manuscript – we have unified the name and used “SCORE” throughout the manuscript

6. Figure 2c - what do the four large circles indicate? Different quadrants? Clusters? – we have removed the circles from the graph as they do not correspond to any type of the analysis. This graph is intended to show the kinase genes with high expression in leukemia and NB cell lines and having high HDAC SCORE at the same time.

7. Line 66 – “chimeric protein” do you mean fusion proteins? – we have corrected this sentence and used “fusion proteins”

8. Figure 1b – indicate what triangle vs. circle mean in the legend – circles stand for leukemia cells lines and triangle – for neuroblastoma, we have updated Figure 1 caption.

9. Lines 216-219 – Or it is because of the limitations of cell lines in recapitulating human disease. The cell lines do change over time because they get used to growing on petri dishes and they accumulate further mutations. -  We agree that the observed differences may be driven by the acquired mutations in cell lines and to make any valuable conclusions more experiments are needed, thus we have removed this sentence from the text in order to prevent misunderstanding of the readers.

10. Lines 231 – “Figure 2a and a” ? – the correct reference is “2a and b”

11. The authors should briefly mention how many HDACs there are and how different/similar they are in the Introduction to make the manuscript more accessible to out-of-field readers. – we have added required information in the Introduction section (Lines 45-50).

References:

  1. Lebedev TD, Vagapova ER, Popenko VI, Leonova OG, Spirin PV, Prassolov VS. Two Receptors, Two Isoforms, Two Cancers: Comprehensive Analysis of KIT and TrkA Expression in Neuroblastoma and Acute Myeloid Leukemia. Front Oncol. 2019 Oct 18;9:1046. doi: 10.3389/fonc.2019.01046. PMID: 31681584; PMCID: PMC6813278.
  2. Lebedev, T., Vagapova, E., Spirin, P. et al. Growth factor signaling predicts therapy resistance mechanisms and defines neuroblastoma subtypes. Oncogene 40, 6258–6272 (2021). https://doi.org/10.1038/s41388-021-02018-7
  3. Rasmussen TA, Schmeltz Søgaard O, Brinkmann C, Wightman F, Lewin SR, Melchjorsen J, Dinarello C, Østergaard L, Tolstrup M. Comparison of HDAC inhibitors in clinical development: effect on HIV production in latently infected cells and T-cell activation. Hum Vaccin Immunother. 2013 May;9(5):993-1001. doi: 10.4161/hv.23800. Epub 2013 Jan 31. PMID: 23370291; PMCID: PMC3899169

Round 2

Reviewer 2 Report

I find the authors' responses to my comments adequate and the current version of the manuscript sufficiently improved. Therefore, I believe the manuscript warrants publication in the Special Issue "The Contribution of Epigenetics to Normal and Malignant Hematopoiesis".